# STT-DPSA: Digital PUF-Based Secure Authentication Using STT-MRAM for the Internet of Things

**DOI:** 10.3390/mi11050502

**Published:** 2020-05-15

**Authors:** Wei-Chen Chien, Yu-Chian Chang, Yao-Tung Tsou, Sy-Yen Kuo, Ching-Ray Chang

**Affiliations:** 1Graduate Institute of Applied Physics, National Taiwan University, Taipei 106, Taiwan; aoowweenn@gmail.com; 2Department of Electrical Engineering, National Taiwan University, Taipei 106, Taiwan; r05943169@ntu.edu.tw (Y.-C.C.); sykuo@cc.ee.ntu.edu.tw (S.-Y.K.); 3Department of Communications Engineering, Feng Chia University, Taichung 407, Taiwan; 4Department of Physics, National Taiwan University, Taipei 106, Taiwan; crchang@phys.ntu.edu.tw

**Keywords:** physical unclonable function (PUF), spin-transfer torque magnetic random-access memory (STT-MRAM), identity authentication, hardware security, internet of things

## Abstract

Physical unclonable function (PUF), a hardware-efficient approach, has drawn a lot of attention in the security research community for exploiting the inevitable manufacturing variability of integrated circuits (IC) as the unique fingerprint of each IC. However, analog PUF is not robust and resistant to environmental conditions. In this paper, we propose a digital PUF-based secure authentication model using the emergent spin-transfer torque magnetic random-access memory (STT-MRAM) PUF (called STT-DPSA for short). STT-DPSA is an original secure identity authentication architecture for Internet of Things (IoT) devices to devise a computationally lightweight authentication architecture which is not susceptible to environmental conditions. Considering hardware security level or cell area, we alternatively build matrix multiplication or stochastic logic operation for our authentication model. To prove the feasibility of our model, the reliability of our PUF is validated via the working windows between temperature interval (−35 ${}^{\circ }$C, 110 ${}^{\circ }$C) and Vdd interval [0.95 V, 1.16 V] and STT-DPSA is implemented with parameters *n* = 32, *i* = *o* = 1024, *k* = 8, and *l* = 2 using FPGA design flow. Under this setting of parameters, an attacker needs to take time complexity O($2^{256}$) to compromise STT-DPSA. We also evaluate STT-DPSA using Synopsys design compiler with TSMC 0.18 um process.

## 1. Introduction

Presently, the pervasiveness of smart devices with embedded memories is affecting our daily life. It raises great opportunities for large-scale applications such as smart city, home automation, etc. In particular, spin-transfer torque magnetic random-access memory (STT-MRAM), an emergent technical term of embedded memory, is expectantly integrated with the internet of things (IoT). That is, the ability of STT-MRAM to operate at low supply voltages, small bit-cell footprint, non-volatility, and high read/write endurance are suitable and attractive for the IoT applications. The IoT physically connected to the STT-MRAM is a new trend technology/application and a well intersection for the development of communication and micro-electromechanical techniques.

Although there are several advantages for integrating STT-MRAM with IoT devices, the issue of security such as device identity authentication for this kind of devices is still an open challenge and thus has drawn much effort for security research communities. More precisely, an unauthenticated or illegitimate device may share fake information to disturb decisions of Cyber-Physical Systems. If the vulnerability on IoT devices is not handled properly, then these emerging concepts potentially expose innumerable data into unsafety. Consequently, identity authentication is in the front line of data protection and can prevent from man-in-the-middle attacks. Unlike the resource-rich server, the computation capability and memory space in typical IoT devices are constrained. Traditional symmetric/asymmetric authentication cryptography architectures are not suitable for IoT devices because they would pose a significant overhead in computation on IoT devices.

Through investigating existed authentication architectures for IoT devices, we classified these architectures into two categories: software and hardware-aided architecture. The software architecture aims to exploit the traditional cryptography algorithm but adjust the strength of the algorithm dynamically to fit the operating requirement of an IoT device. The architectures usually suffers from heavy computation of the cryptography algorithm and terribly consumes the usage of memory space. In particular, they are vulnerable to hardware cloning attacks. In contrast, another solution is hardware-aided architecture, e.g., Physical Unclonable Function (PUF)-based architecture [1]. PUF is a new technology exploiting the manufacturing variability of Integrated Circuit (IC) as the unique fingerprint of each IC, in which the input to the PUF is referred as the “*challenge*” and the output is referred as the “*response*”. Therefore, a PUF can be modeled as a set of *challenge-response* pairs (CRPs). Briefly, the PUF, a physical entity, can produce output value that is dependent of its physical structure to make it hard to be cloned. With the PUF, researchers have a chance to develop ultra-fast and efficient security primitives against side-channel and other common physical attacks [2].

PUF can be classified into two types [3], namely weak PUF and strong PUF. The weak PUF (resp. the strong PUF) means that the number of CRPs is linearly (resp. exponentially) in accordance with the number of components whose behavior depends on manufacturing variation. PUF that is a hardware-aided architecture aims to make the computation efficient can reduce the usage of memory space and resist hardware cloning attacks. However, the hardware-aided architecture has a fatal weakness in being susceptible to environmental conditions.

Our main contributions are summarized as follows:We propose a digital PUF-based secure authentication model using STT-MRAM PUF as configuration bits for the network of LUTs.We use two digital PUFs that have the same function as the basic building block for IoT authentication models.We present two versions of authentication model for different requirements on cell area: one is based on matrix multiplication and the other is based on stochastic logic.Through a series of attack analyses and hardware resource evaluations, we prove that our method poses lightweight overhead on participating parties and provides high security for IoT devices.We implement STT-DPSA on FPGA to prove the feasibility of our model.

The rest of this paper is organized as follows. Section 2 describes related works. Section 3 introduces our system model and attack model. We then describe the details of proposed methods in Section 4 and Section 5. We give a system analysis including system complexity and security in Section 6, followed by the experimental results in Section 7. Finally, the conclusion is presented in Section 8.

## 2. Related Work

To address the essential security problem of identity authentication for low-end IoT devices, it takes much effort in the research of software-based and hardware-based authentication.

Aiming to software-based authentication, the research community in sensor networks strives to devise lightweight and practically feasible but computationally less intensive cryptography algorithms for the resource-limited hardware with lower computation capability, smaller memory size, and limited energy supply. Some studies based on symmetric-key cryptography, such as SPINS [4], TinySec [5], LEAP [6] and SER [7], have taken less computation cost and been proven to be feasible on IoT devices. However, their methods incur a large communication overhead for key distribution. As for public key cryptography, TinyECC [8] and TinyPBC [9] have been proposed to tackle the expensive key computation cost using key size reduction. Although [4,5,6,7,10] have taken effort to enhance the security of IoT devices, they still suffer from a dilemma of computation overhead and security level. The purely software-based cryptography methods suffer from inefficient computations, dramatically consumption of memory space, and have the critical vulnerability of hardware cloning attacks under the inherent resource-constrained environment.

From the perspective of hardware-aided authentication model, PUF is attractive to the research realm recently. Layman et al. [11] proposed a weak PUF, namely Static Random-Access Memory (SRAM) PUF, consisting of symmetric and cross-coupled inverters. Once the power of the SRAM PUF is on, manufacturing variability will be leveraged to push each cell of SRAM PUF toward the logical 1 or 0. Notably, SRAM PUF exploits only one CRP for cryptography key generation rather than device authentication, in which *response* is an unknown initial state caused by manufacturing variability.

Gassend et al. [2] proposed a type of strong PUF, namely Arbiter PUF, to generate several ideal identical delay paths and apply multiplexers to switch between these paths. To create ideal identical delay paths, Arbiter PUF is designed in the layout level. Moreover, the *challenge* to the Arbiter PUF is the control signal of multiplexers. With implementing the Arbiter PUF, two ideal identical paths are fed into an arbiter [12]. When the input of the Arbiter PUF changes, the signal is propagated through two different ideal identical delay paths. Because manufacturing variability of different hardware, the arrival time of these two input signals in an arbiter will be various. As a result, the arbiter can generate a *response* depended on which signal arrives first. This type of PUF can generate several CRPs and be used for lightweight authentication model on IoT devices. The typical deployment of strong PUF for identity authentication model is shown in Figure 1 and described as follows.

The client side equipped with a PUF is manufactured with several CRPs. Moreover, the verifier side has a database of the client’s CRPs. When the client needs to be authenticated, the verifier randomly selects one CRP from the database. Then, the verifier sends a selected *challenge* to the client and reserves the corresponding *response*
*b* on his/her hand. Once the client receives the *challenge*, the *challenge* is fed into a PUF for generating a corresponding *response*
$b{}^{\prime }$. Subsequently, the client sends the $b{}^{\prime }$ to the verifier. Finally, the verifier determines that if $b=b{}^{\prime }$, the client is legitimate; otherwise, the client fails to be authenticated.

However, a strong PUF has exponentially growth numbers of CRPs with respect to components whose behaviors are depended on manufacturing variation. This makes the verifier cannot allocate sufficient memory space for entire CRPs of the strong PUF. Additionally, if a CRP is used repeatedly in an authentication model, an attacker can easily crack the authentication model by sending the used *response* to the verifier. As for computationally efficiency, Matched Public PUF (PPUF) [13] was proposed in leveraging the circuit aging model to produce an identical PUF. Nevertheless, it needs high implementation requirement in terms of measurement accuracy and environmental stability.

Xu et al. [14] first proposed digital PUF architecture to resolve the problem of environmental susceptibility in [2,11,13]. They developed Digital Bimodal Function (DBF) by using a randomly generated Boolean function as the large-scale network of lookup tables (LUTs) and exploiting the irreversibility of decomposition of Boolean function to represent the same Boolean function in two forms, namely compact form and complex form. The method in [14] was implemented on FPGA and aimed to exploit the huge gap of computation time and hardware resource between these two forms. However, when configuration bits of LUTs are compromised, the DBF is easily cloned. Therefore, Xu et al. [15] further proposed an advanced model by using Arbiter PUF to generate configuration bits for the network of LUTs. Because the variation of environmental conditions, only stable CRPs can be used to configure the network of LUTs. To provide a practical usage, the Arbiter PUF must take additional effort to obtain stable CRPs under different environment conditions. Some hardware-aided architectures, such as [16,17,18,19,20], also have a fatal weakness in being susceptible to environmental conditions.

In 2015, Xu et al. [21] proposed a digital PUF approach by observing intentional faults that change the function of IC dramatically. Subsequently, Miao et al. [22] proposed a digital PUF approach by using variation of lithography. The digital PUF proposed in [21] was hard to be applied in real-world applications because the intentional faults not only affect the function of IC, but also affect the operating region of transistors. The digital PUF [21] might cause much power consumption and put Complementary Metal-Oxide-Semiconductor (CMOS). gates into an uncertain operating region. In contrast to [21], the digital PUF proposed in [22] was only implemented in simulation stage. The digital PUF in [21] and [22] were difficultly duplicate in both client and verifier sides to drive identity authentication while the digital PUF proposed in [14] can be applied easily. Worse, a typical strong PUF-based authentication model will consume unacceptable memory usages for the verifier.

To solve the drawback of unacceptable memory usages, an emerging method called Public PUF (PPUF) was proposed by [23]. As for the PPUF-based model, compared to the need for the internal secret storage of CRPs, the PPUF model has no secret information. In other words, the authentication information is stored publicly instead of being stored in the verifier side. As a result, this type of PUF does not consume a large amount of memory.

Park et al. [24] proposed a PPUF-based authentication model, namely PUFSec, exploiting the gap of *response* generation time in the PPUF model and PPUF hardware. The PPUF hardware can compute the *response* in a measurably faster time compared to PPUF model. That is, the PPUF-based method exploits a huge computation gap between simulation and execution, and computation gap of challenge between verifier and attacker. However, the computation complexity of the participating parties in PUFSec would growth exponentially with respect to the size of PUF hardware area. Moreover, selecting an appropriate $T{}^{\prime }$ which is the threshold time in the PPUF-based model is an open challenge in computationally efficiency for the distinction of an attacker and a legitimate device.

A new low-cost PUF-based temperature sensor for secure remote temperature sensing was proposed by Cao et al. [25]. Cao et al. exploited the approximately linear positive temperature coefficient of CMOS inverter in super-threshold operation to calibrate the running frequency of ring oscillator in a reconfigurable ring oscillator PUF at different temperature. The ring oscillator frequency corresponding to the sensed temperature is fed into a randomizer seeded by the input challenge to select new ring oscillator pairs for comparison to generate a random, unique and physically unclonable digital tag, which is valid for a selected input challenge to a target device at a particular temperature. In 2018, Gu et al. [26] proposed the first IoT secure communication framework for BLE-based networks that guards against device spoofing via fingerprinting-based device authentication. These designs are orthogonal to our STT-MRAM-based PUF.

In this paper, we propose a digital PUF-based authentication model using STT-MRAM PUF for IoT devices. To address the problems of exponential initialization time and dramatically memory consumption caused in [14,15], we use two identical digital PUF for our authentication model. Moreover, we provide the configuration bits of LUTs using STT-MRAM PUF. Because the hardware area of STT-MRAM PUF is significantly lower than others PUF such as Arbiter PUF and SRAM PUF [27], we use STT-MRAM PUF to configure the network of LUTs. The advantages of our digital PUF-based authentication model include efficient computation, efficient memory usages, and unclonable hardware using stable CRPs of STT-MRAM PUF regardless of operational and environmental conditions. The computation overhead of participating parties in our protocol is lightweight while our model still exploits the huge gap between execution and simulation, so that an attacker needs to take much effort in terms of execution time and computational complexity for compromising our architecture.

To the best of our knowledge, we first propose a method using two digital PUFs with the cell of STT-MRAM PUF, which have the same function as the basic building block for the IoT authentication architecture. Table 1 presents comparisons of our model with previous studies in terms of the computational complexity, susceptibility, initialization time, and the resource complexity. Our method outperforms the well-known methods in terms of the computational and resource complexity but not susceptible to environmental conditions.

## 3. System and Attack Models

In this section, we describe the system model and attack model for our proposed architecture.

### 3.1. System Model

Our system model is shown in Figure 2. The PUF circuit used in STT-DPSA is a digital model and a type of strong PUF. STT-DPSA has numerous CRPs with the ideal performance of statistic metrics, i.e., inter-Hamming distance being equal to 0 and uniformity of *response* being equal to 0.5 [28]. For easy representation, the design of PUF in a digital model for STT-DPSA and the design of STT-MRAM PUF are represented as $P^{D}$ and $P^{S}$, respectively.

In addition, we assume that there are two roles in our model, namely verifier and client. The verifier, a moderator in an IoT network, is a trusted part for authenticating clients via $P^{D}$. The client, an IoT device, intents to join a dominated network for communicating with other legitimate members.

### 3.2. Attack Model

We assume that attackers can eavesdrop the information transmitted between the verifier and client, but attackers cannot retrieve any information stored in both of them. In other words, attackers can perform man-in-the-middle attacks through recording the information exchanged between the verifier and client, and try to impersonate the legitimate client in the next round of authentication based on these records.

## 4. Design of a Strong Digital Physical Unclonable Function (PUF) Based on Spin-Transfer Torque Magnetic Random-Access Memory (STT-MRAM) PUF

The structure of $P^{D}$ is shown in Figure 3, which composes of several LUTs that are configured by $P^{S}$ to mitigate environmental susceptibility.

A cell of STT-MRAM and $P^{S}$ used in our method are shown in Figure 4 and Figure 5, in which the MTJ is a component consisting of two ferromagnetic layers separated by a thin insulator and is implemented by two states using simple resistor model: anti-parallel (AP) and parallel (*P*). The resistance of the MTJ in the *P* state is comparably lower than the resistance of the MTJ in the AP state. In theory, if the two MTJs have the same state and all their physical parameters are the same, then they should have the same resistance value. In fact, they have manufacturing variations led to the inevitable difference between the two ideal identical resistance values. As a result, the $P^{S}$ exploits the resistance difference caused by the manufacturing variability of the MTJs in the same state. The dual in-line package of our MTJs is shown in Figure 6. Through wire bonding, two adjacent MTJs are selected to use.

$P^{S}$ uses two cross-coupled inverters to amplify the difference between the resistances of the MTJs in the same state. Figure 5 exhibits the design of $P^{S}$, in which output signal BIT is a *response* of the $P^{S}$. The generation process of *response* for the $P^{S}$ is as follows.

In the beginning, MTJ A and MTJ B are configured to the same state. Active-low signal RCLK is set to high voltage while the two NMOS transistors controlled by RCLK is ‘on’ and the top PMOS transistor connected to Vdd is ‘off’. Moreover, BIT and BITB are short to the digital ground. After switching RCLK to low voltage, the top PMOS transistor is ‘on’ and the two NMOS transistors controlled by RCLK become ‘off’. BIT and BITB are pre-charged and activate the other transistors. Since the two cross-coupled inverters are activated, BIT and BITB are amplified to opposite digital levels in the regeneration behavior by the resistance difference between MTJ A and MTJ B. At the end, the output signal BIT is a *response* of the $P^{S}$ and BITB is the inverse of BIT.

Notably, the $P^{S}$ is used as a key of LUTs. That is, we exploit the $P^{S}$ to generate configuration bits of LUTs. Therefore, we just use the *response* of a $P^{S}$ once. The validation of unpredictability for a $P^{S}$ under different environmental conditions is demonstrated in Section 7. With the composition of our $P^{S}$ and LUTs, we can model the $P^{D}$ in our authentication architecture.

We decompose $P^{D}$ into multiple stages, in which one stage consists of several LUTs. Let the number of primary input be *i* and the number of primary output be *o*. In our design, we let i=o; let the number of stages be *l*; let the number of LUTs be equal to *i* in each stage of the $P^{D}$. As shown in Figure 3, $P_{0}^{D}\left(0:i-1\right)$ is a bus consisting *i* primary inputs and is the input for the first stage of the LUT network when the output for the first stage is $P_{1}^{D}\left(0:i-1\right)$. Each stage of $P^{D}$ follows the same rule. Finally, $P_{l}^{D}\left(0:i-1\right)$ will be the bus that consists of *i* primary outputs. Each LUT is programmed using verilog code in the structure of 4-inputs LUT, as exhibited in the following verilog code of module LUT (in, value), in which the signal in (3:0) is the input of LUT module consists of 4 bits while the signal value is the output bit of a LUT module.

Notably, the configuration bits in the always statement of the verilog code must be provided in an unpredictable manner. The response signal from BIT of a $P^{S}$ is used to configure the LUT. For example, 16 cells of $P^{S}$ are used to configure a 4-input LUT, in which the response signals (i.e., BITs) are 1110000100000001. Next, we describe our authentication model based on the design of $P^{D}$ using $P^{S}$ as configuration bits.

## 5. Authentication Model

In this section, we present two authentication models. One is based on matrix multiplication and the other is based on stochastic logic, in which both of them exploit $P^{D}$ as a common building block.

### 5.1. Authentication Model Based on Matrix Multiplication

To develop an efficient and a feasible PUF-based authentication model in terms of computational time and memory space regardless of operational and environmental conditions, we do not directly exchange CRPs generated by $P^{D}$ between the client and verifier for identity authentication. Instead, we propose a strong digital PUF-based authentication model through exchanging the CRPs of matrix *A* that will be updated in each authentication round using the *response* of $P^{D}$ between the verifier and client. Our authentication model based on matrix multiplication is shown in Figure 7.

Before authentication, we must deploy the matrix *A* and $P^{D}$ to the verifier and client initially. Using the deployed matrix *A* and $P^{D}$, behaviors of the verifier and client are as follows.
**Step 1:** The verifier (resp. the client) generates a random number $N_{v}$ (resp.$N_{c}$) using pseudo-random number generator (PRNG) [12] and exchanges the random number with the client (resp. the verifier).**Step 2:** Both of the verifier and client form a string *C* by concatenating $N_{c}$ with $N_{v}$.**Step 3:** Both of the verifier and client use the string *C* as a *challenge* of his/her $P^{D}$ and obtain a corresponding *response* (called *R* for short) in a matrix form.**Step 4:** Both of the verifier and client compute the matrix multiplication AR and replace matrix *A* with AR. If the resulted matrix *A* is a zero matrix, then we must re-initialize the matrix *A* and go back to step 1. That is because a zero matrix used in step 5 will lead to a result of zero vector that can be easily predicted by an attacker.**Step 5:** The verifier selects a column vector *x* generated by PRNG as a *challenge* and sends *x* to the client. Subsequently, the verifier (resp. the client) computes Ax as *b* (resp.$b{}^{\prime }$). Next, the client sends $b{}^{\prime }$ to the verifier. If $b=b{}^{\prime }$, then the verifier authorizes the client as a legitimate device. Otherwise, the client fails to be authorized and viewed as an illegitimate one. We show behaviors of the verifier and client in Algorithms 1 and 2, respectively.

**Algorithm 1:** The verifier Side**Input**: PD, *A*, Nc, and b′**Output**: Whether the authentication process succeeds// Flow chart of authentication model based on matrix multiplication for the verifier side is as shown in the left side of Figure 8.
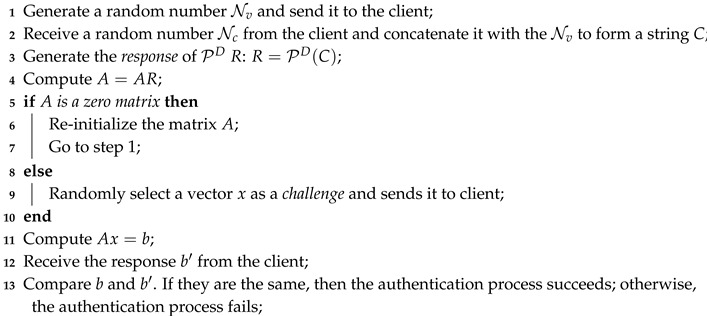


**Algorithm 2:** The client Side**Input**: PD, *A*, Nv, and *x***Output**: b′// Flow chart of authentication model based on matrix multiplication for the client side is as shown in the right side of Figure 8.
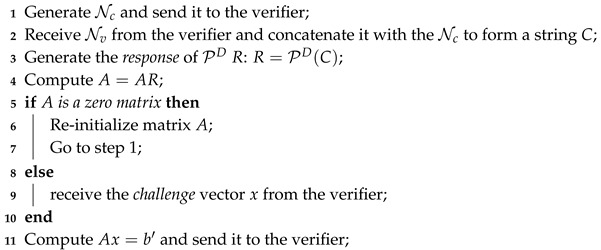


Notably, we exchange the vector *x* and the vector *b* between the verifier and client. These two vectors are the information regarding to the matrix *A*. More precisely, the vector *x* is an input of *A*. After operating a matrix multiplication, the vector *b* becomes an output of *A*. Therefore, we call the exchanged vectors as CRPs of *A*. For easy presentation, the CRPs of *A* is known as $CRP^{A}$. Because we refresh the *A* at each round of authentication using the *response* of $P^{D}$, it is difficult to predict the next round of $CRP^{A}$ based on the used one. Therefore, $P^{D}$ can satisfy the requirement of designing a strong PUF-based authentication model [3].

### 5.2. Authentication Model Based on Stochastic Logic

Before introducing the authentication model based on stochastic logic, we present the preliminary knowledge of stochastic logic used to construct our authentication model. Unlike the traditional binary encoding used to treat data in binary number, stochastic logic is applied to encode the data in the form of a bitstream using probability model [29,30], in which the probability for a bitstream with *r* 1’s and N−r 0’s denotes r/N. With operating stochastic logic, addition and multiplication are candidates that can be processed using only one logic component.

Using the stochastic adder, the two inputs of adder are bitstreams $x_{b}$ and $y_{b}$, and the output is the bitstream $z_{b}$ while a random input *s* generates 0 or 1 with fair probability. The vectors *x* and *y* are encoded as the probability of the specific bit when the bitstreams $x_{b}$ and $y_{b}$ are equal to 1. In one clock cycle, the probability of the specific bit when $z_{b}=1$ is $\frac{1}{2}\times x+\frac{1}{2}\times y$. Consequently, if the length of bitstreams $x_{b}$ and $y_{b}$ is *N*, then the expected number of bit 1 in $z_{b}$ is $\frac{x+y}{2}$. This is a scaled adder of two values implemented by one multiplexer. Using a multiplier with two inputs implemented by one AND gate, the two inputs of AND gate are bitstreams $x_{b}$ and $y_{b}$, and the outputs are the bitstreams $z_{b}$. The vectors *x* and *y* are encoded as the probability of the specific bit in bitstream $x_{b}$ being 1. Assumed that the encoding bitstreams $x_{b}$ and $y_{b}$ are independent. In one clock cycle, the probability of the specific bit when $z_{b}=1$ is x×y. Consequently, if the length of bitstreams $x_{b}$ and $y_{b}$ is *N*, then the expected number of bit 1 in $z_{b}$ is x×y. Generally, the cell area for designing stochastic logic is significantly small. Therefore, we synthesize stochastic logic adder and multiplier to reduce the cell area of our model. Subsequently, we will present our authentication model based on stochastic logic.

The authentication model based on stochastic logic is shown in Figure 9, in which $P^{D}$ is used as a building block. Unlike authentication model based on matrix multiplication, stochastic logic version does not need to initialize the matrix *A* in both verifier and client. Using the deployed $P^{D}$, behaviors of the verifier and client are as follows.
**Step 1:** The verifier (resp. the client) generates a random number $N_{v}$ (resp.$N_{c}$) using PRNG and exchanges the random number with the client (resp. the verifier).**Step 2:** The verifier and client form a string $C_{1}$ by concatenating $N_{c}$ with $N_{v}$.**Step 3:** The verifier and client use the string $C_{1}$ as a *challenge* of his/her $P^{D}$ and obtain a corresponding *response* (called $R_{1}$ for short).**Step 4:** The verifier and client repeat steps 1−3 and obtain a corresponding *response* (called $R_{2}$ for short).**Step 5:** The verifier and client treat $R_{1}$ and $R_{2}$ as bitstreams and take these two bitstreams as inputs to a stochastic logic adder for obtaining a corresponding calculation result value and $value^{{}^{\prime }}$.**Step 6:** The client sends $value^{{}^{\prime }}$ to the verifier. Finally, if $value=value^{{}^{\prime }}$, then the verifier authorizes the client as a legitimate device. Otherwise, the client fails to be authorized and viewed as an illegitimate one. We show the behaviors of the verifier and client in Algorithms 3 and 4, respectively.

**Algorithm 3:** Verifier Side**Input**: PD, Nc, value, and value′**Output**: Whether the authentication process succeeds// Flow chart of the authentication model based on stochastic logic for the verifier side is as shown in the left side of Figure 10.1Generate a random number Nv and send it to the client;2Receive a random number Nc from the client and concatenate it with the Nv to form a string C1;3Generate the *response* of PD
R1: R1=PD(C1);4Repeat steps 1–3 and obtain R2=PD(C2);5Treat R1 and R2 as bitstreams and compute value=(R1+R2)/2 using stochastic logic adder;6Receive the value′ from the client;7Compare value with value′. If value=value′, then the authentication process succeeds;otherwise, the authentication process fails;

**Algorithm 4:** Client Side**Input**: PD, and Nv**Output**: value′// Flow chart of the authentication model based on stochastic logic for the client side is as shown in the right side of Figure 10.1Generate a random number Nc and send it to the verifier;2Receive a random number Nv from the verifier and concatenate it with the Nc to form a string C1;3Generate the *response* of PD
R1: R1=PD(C1);4Repeat steps 1–3 and obtain R2=PD(C2);5Treat R1 and R2 as bitstreams and compute value′=(R1+R2)/2 using stochastic logic adder;6Send the value′ to the verifier;

Notably, in the step 5 of authentication model, the selected input of stochastic logic multiplexer is provided by the same random source deployed in the verifier and client. With this assumption, value is equal to $value{}^{\prime }$ if they are transmitted over the air without being destroyed by an attacker. In the next section, we will analyze our proposed model in system complexity and security.

## 6. System Analysis

Because our proposed model is implemented in digital circuit, two major issues need to be discussed. First, the cell area of digital circuit affects the cost of manufacturing. Second, the delay of critical path affects the upper bound of operating frequency. Therefore, we analyze the cell area and the delay of critical path of proposed model based on ASIC cell-based design flow and prove that our method can resist against man-in-the-middle attacks in the following sections.

### 6.1. System Complexity

The dimension of the matrix *A* is n×n, in which each element in *A* is represented by *k* bits. Let the length of input bits of $P^{D}$ be *i* and let the length of output bits of $P^{D}$ be *o*, where i=o. Moreover, a 4-input LUT is used as a basic block for $P^{D}$. We also let the length of $P^{D}$ be *l*. Subsequently, we will discuss the effects of parameters *n*, *k*, and *l* on the cell area and the delay of critical path for our authentication model.

In the first three steps of matrix multiplication process (as shown in Figure 7), $N_{v}$ and $N_{c}$ are generated using a PRNG and exchanged between the client and verifier to produce a $challenge_{p}$*C*. Next, *C* is used as an input for $P^{D}$ to produce a (n×n)-matrix *response R* which will be multiplied by the matrix *A*. Finally, the client and verifier execute matrix multiplications to do identity authentication. For estimating the cell area of the proposed model, we evaluate the effect of matrix multiplications and the execution of $P^{D}$. Because the cell area of matrix multiplication for two n×n matrices is significantly larger than that of matrix multiplication for one n×n matrix by one n×1 matrix, we only evaluate the cell area of matrix multiplication for two n×n matrices. In addition, the cell area of PRNG is smaller than the cell area of matrix multiplication. As for the delay of critical path, it is mainly incurred by matrix multiplications and the execution of $P^{D}$. The delay of matrix multiplication for two n×n matrices is higher than the matrix multiplication for one n×n matrix by one n×1 matrix. In terms of PRNG, LFSR is implemented as a PRNG. When the clock edge is triggered, the PRNG is ready to be used. For simplicity of analysis, we put the execution of $P^{D}$ and the matrix multiplication of two n×n matrices in the same clock cycle. Operations of other parts in proposed model are put in other clock cycles. By this approach, we make sure that the execution of $P^{D}$ and the matrix multiplication of two n×n matrices are the critical paths.

As shown in Figure 9, the first three steps are as the same as matrix multiplication. The step 4 is just the repeat of steps 1–3. To compute value and $value{}^{\prime }$, we only need one multiplexer to compute. Consequently, the cell area and the delay of critical path for one multiplexer are significantly lower than that for matrix multiplication. We assume that the delay of critical path and cell area of stochastic logic version for authentication model mainly come from $P^{D}$. In the following, we analyze the cell area and the delay of critical path in the verifier and client for matrix multiplication version and stochastic logic version.

**Theorem** **1.**
*When authentication model is based on matrix multiplication, the cell area at the verifier/client side is formulated as:*
(1)$$\begin{array}{c} A_{c}=i\cdot l\cdot LUT_{c}+n^{3}\cdot k\cdot A_{am}+R, \end{array}$$
*where i, l, $LUT_{c}$, n, k, $A_{am}$, and R denote the number of bits input to $P^{D}$, the length of $P^{D}$, the average cell area of a 4-input LUT, the dimension of the matrix A, the bits of each element in the matrix A, the average cell area of 1-bit adder and multiplier, and the others, respectively.*


**Proof.** The two main factors affecting the cell area at the verifier/client side are the execution of $P^{D}$ and matrix multiplication, in which the number of LUTs in the $P^{D}$ is i·l. As a result, the cell area of $P^{D}$ is $i\cdot l\cdot LUT_{c}$, where $LUT_{c}$ is the average cell area of a 4-input LUT. For the matrix multiplication, each element of new matrix generated by the multiplication of two n×n matrices is the result of inner product of two n×1 vectors, where inner product operation of two n×1 vectors needs *n* adders and *n* multipliers. A new matrix generated by two n×n matrices has $n^{2}$ elements. Therefore, multiplication of two n×n matrices needs n·n·n adders and multipliers. As for the parameter *k*, the number of bits used to encode one element of *A* has linear effect on the cell area of the adder and multiplier. Taking operation of addition of two 3-bit values as an example, addition of two 3-bit values will need 15 gates in a typical case. When we change the number of bits to encode one element of *A*, it has linear effect on the cell area of an adder. The operation of a multiplication is the same as that of an adder. As a result, the cell area of matrix multiplication is $n^{3}\cdot k\cdot A_{am}$, where $A_{am}$ is the average cell area of 1-bit adder and 1-bit multiplier. Finally, we represent the area of the remaining part of proposed model (i.e., PRNG) as *R*. Therefore, the cell area at the verifier/client side is $i\cdot l\cdot LUT_{c}+n^{3}\cdot k\cdot A_{am}+R$. □

**Theorem** **2.**
*When authentication model is based on matrix multiplication, the delay of critical path at the verifier/client side is formulated as:*
(2)$$\begin{array}{c} D_{cp}=l\cdot LUT_{d}+n\cdot k\cdot A_{DD}, \end{array}$$
*where n, k, $A_{DD}$, l, and $LUT_{d}$ denote the dimension of the matrix A, the bits of each element in the matrix A, the average delay of a 1-bit adder, the length of $P^{D}$, and the average delay of a 4-input LUT, respectively.*


**Proof.** The $D_{cp}$ for the verifier/client side is incurred by the execution of $P^{D}$ and the matrix multiplication AR. From the primary input of $P^{D}$ to the output of $P^{D}$, as shown in Figure 3, the parallel network of LUTs is with *l* levels of 4-input LUTs. Therefore, $D_{cp}$ is proportional to the length of $P^{D}$ and represents as $l\times LUT_{d}$. Additionally, to analyze the execution of matrix multiplication, multiplication of two n×n matrices will generate a new matrix with $n^{2}$ elements, where each element is the result of inner product of two n×1 vectors. Moreover, $D_{cp}$ is affected by the delay of executing an adder while processing inner product for these two matrices to result in $n^{2}$ elements. These resulted elements can be computed completely within the delay time of executing *n* adders. As for the parameter *k*, it has the linearly effect on the delay of matrix multiplication. Taking an operation for adding two 3-bit values, the computation time for the addition of these two values is in 3-adder delay. When we change the number of bits to encode one element of the matrix *A*, the number of bits used to encode one element of the matrix *A* has linearly effect on the delay of matrix multiplication. That is, the delay of matrix multiplication can be formulated as $n\times k\times A_{DD}$. Therefore, the delay of critical path at the verifier/client side is $l\cdot LUT_{d}+n\cdot k\cdot A_{DD}$. □

**Theorem** **3.**
*When authentication model is based on stochastic logic, the cell area at the verifier/client side is formulated as:*
(3)$$\begin{array}{c} A_{c}=i\cdot l\cdot LUT_{c}+R, \end{array}$$
*where i, l, $LUT_{c}$, and R denote the number of bits input to $P^{D}$, the length of $P^{D}$, the average cell area of a 4-input LUT, and the others, respectively.*


**Proof.** The main factor affecting the cell area in the verifier/client side are the execution of $P^{D}$, in which the number of LUTs in $P^{D}$ is i·l. As a result, the cell area of $P^{D}$ is $i\cdot l\cdot LUT_{c}$, where $LUT_{c}$ is the average cell area of a 4-input LUT. Finally, we represent the cell area of the remaining part (i.e., PRNG) as *R*. Therefore, the cell area in the verifier/client side is $i\cdot l\cdot LUT_{c}+R$. Obviously, the cell area in using stochastic logic for authentication model is smaller than the cell area in using matrix multiplication for authentication model. □

**Theorem** **4.**
*When authentication model is based on stochastic logic, the delay of critical path at the verifier/client side is formulated as:*
(4)$$\begin{array}{c} D_{cp}=l\cdot LUT_{d}, \end{array}$$
*where l and $LUT_{d}$ denote the length of $P^{D}$ and the average delay of a 4-input LUT, respectively.*


**Proof.** $D_{cp}$ for the verifier/client side is incurred by the execution of $P^{D}$. From the primary input of $P^{D}$ to the output of $P^{D}$, as shown in Figure 3, the parallel network of LUTs is with *l* levels of a 4-input LUT. Because $D_{cp}$ is proportional to the length of $P^{D}$ and represents as $l\times LUT_{d}$, the delay of critical path at the verifier/client side is $l\cdot LUT_{d}$. Obviously, the delay of critical path in using stochastic logic is smaller than that using matrix multiplication for authentication model. □

In summary, our proposed model in matrix multiplication and stochastic logic poses linearly growth on delay of critical path and poses polynomial growth on cell area when changing our system parameters. More importantly, our model is not affected by environmental condition because it is implemented using digital circuit.

### 6.2. System Security

In this section, we prove that our system can resist against man-in-the-middle attacks, in which an attacker may try to break our system by eavesdropping information exchanged between the verifier and client. The attacker would drive the following attacks: (1) $P^{D}$ modeling attacks, (2) modeling attack to the matrix *A* of matrix multiplication authentication model, (3) brute-force attacks to $P^{D}$, (4) brute-force attacks to the matrix *A* of matrix multiplication authentication model, (5) brute-force attacks to the *response* of the matrix *A* of matrix multiplication authentication model, (6) value modeling attacks to stochastic logic authentication model, and (7) resistance against machine-learning attacks.

#### 6.2.1. $P^{D}$ Modeling Attacks

The $P^{D}$ modeling attack is a machine-learning type of attacks [31], which aims to predict the behaviors of $P^{D}$ by observing some CRPs of $P^{D}$. In fact, we do not reveal any parameter of $CRP_{D}$ in our authentication model. As a result, our authentication model is resilient against $P^{D}$ modeling attacks.

#### 6.2.2. Modeling Attack to the Matrix *A* of Matrix Multiplication Authentication Model

An attacker may intent to eavesdrop the information transmitted over the air and try to model the matrix *A* to compute the privacy parameter *b*.

**Lemma** **1.**
*If the same matrix A exists in different authentication rounds, then the matrix A can be easily obtained by solving the linear equation with the known CRPs.*


**Proof.** Let the dimension of the matrix *A* be n×n. In our authentication model, we resolve the same linear system Ax=b between the verifier and client, where the dimensions of *x* and *b* are n×1. During the process of authentication, the model will reveal a pair (*x*, *b*). For example, if a legitimate client successfully gets the authentication from the verifier three times, then an attacker can resolve the matrix *A* by using the pairs ($x_{1}$, $b_{1}$), ($x_{2}$, $b_{2}$), and ($x_{3}$, $b_{3}$). The attacker can form a new linear system $A{}^{\prime }x=b$, where the dimension of $A{}^{\prime }$ is $\left(3*n\right)\times n^{2}$ and the dimensions of *x* and *b* are $n^{2}\times 1$. Expanding this concept, if the client successfully gets the authentication from the verifier *j* times, then the attacker can form a new linear system $A{}^{\prime }x=b$, where the dimension of $A{}^{\prime }$ is $\left(j*n\right)\times n^{2}$ and the dimension of *x* and *b* are $n^{2}\times 1$. In this linear system, it satisfies: $rank\left(A{}^{\prime }\right)\leq n^{2}<\left(j*n\right)$. If we randomly select *challenges* in the model with increasing authentication rounds, then the rank of $A{}^{\prime }$ constructed by an attacker will increase. Finally, the rank of $A{}^{\prime }$ will be the upper-bound value $n^{2}$. The solutions of this linear system include the general solution of $A{}^{\prime }x=0$ and a particular solution of $A{}^{\prime }x=b$. If the matrix $A{}^{\prime }$ is full column rank, then the number of solution will be zero or one solution. Moreover, we know that there must be a solution that is exactly the matrix *A* in the linear system. The attacker can rule out the case of no solution and solve the linear system. As a result, the attacker can solve the matrix *A* easily after several rounds of authentication. □

In the authentication model based on matrix multiplication, we refresh the matrix *A* in every rounds of authentication. For an attacker, he/she needs to obtain *A* at one specific time by observing one $CRP^{A}$. However, to obtain the matrix *A* by observing one $CRP^{A}$ results in a linear system of infinitely many solutions.

**Lemma** **2.**
*If we refresh the matrix A for every authentication rounds, then A will be infinitely many solutions for the linear system constructed by an attacker.*


**Proof.** Because we refresh the matrix *A* for every rounds of authentication, the linear system constructed by the attacker actually results different matrices in every rounds of authentication. Moreover, these matrices are mutually independent, so the attacker can only resolve the matrix *A* at ending of one specific round by constructing the linear system using one CRP. Suppose that the number of elements in the matrix $\hat{A}$ is $n\times n^{2}$. Holding $rank\left(\hat{A}\right)\leq n<n^{2}$, the number of elements in null space of the matrix $\hat{A}$ is $n^{2}-rank\left(\hat{A}\right)>0$. The solutions of linear system are the general solution of $\hat{A}x=0$ and one particular solution of $\hat{A}x=\hat{b}$. This linear system must have one solution that is exactly the matrix *A*, thus, the attacker can rule out the scenario of no solution. Consequently, the linear system $\hat{A}x=\hat{b}$ constructed by the attacker has infinitely many solutions. □

Briefly, it is hard to model the matrix *A* or $P^{D}$ using the recorded information. However, an attacker may try to use brute-force attacks against the matrix *A* or the $P^{D}$. Therefore, we analyze brute-force attacks to the $P^{D}$ and the matrix *A* in the following sections.

#### 6.2.3. Brute-Force Attacks to $P^{D}$

As proven in Section 6.2.1, an attacker cannot model the $P^{D}$ effectively using the recorded information. However, the attacker can blindly guess the configuration bits of the $P^{D}$. In other words, the attacker can try to duplicate the $P^{D}$ using brute-force methods. Given l·i LUTs in the $P^{D}$, we need to use 16·l·i bits to model the $P^{D}$ in the initialize phase of proposed model. As a result, the attacker will take exponentially time complexity $O(2^{16\cdot l\cdot i})$ to compromise the $P^{D}$. In contrast, $D_{cp}$ of our model only costs linear time complexity when the $P^{D}$ and the matrix *A* are known in system.

We can easily create huge computational gap between our proposed model and the knowledge of an attacker using brute-force attacks to model $P^{D}$. Even if the attacker can duplicate the $P^{D}$, he/she still needs to compromise the matrix *A* to break our model. Therefore, we analyze the system security when an attacker exploits brute-force attacks to compromise the matrix *A* in the next section.

#### 6.2.4. Brute-Force Attacks to the Matrix *A* of Matrix Multiplication Authentication Model

With performing man-in-the-meddle attacks, an attacker needs to find one solution among infinitely many solutions of a linear system to model the matrix *A*. However, the attacker can only blindly guess elements in the matrix *A*. In other words, the attacker can only try to duplicate the matrix *A* using brute-force methods.

Let the number of elements in the matrix *A* be n×n and each element in the matrix *A* be *k* bits. The computational complexity to model the matrix *A* is exponential complexity $O(2^{k\cdot n^{2}})$. In contrast, the complexity of $D_{cp}$ is linear with $n\cdot k\cdot A_{DD}+l\cdot LUT_{d}$.

After discussing the brute-force attacks against $P^{D}$ and the matrix *A*, an attacker can also focus on breaching the *response* of the matrix *A*. Consequently, we analyze brute-force attacks to the *response* of the matrix *A* in the next section.

#### 6.2.5. Brute-Force Attacks to the *Response* of the Matrix *A* of Matrix Multiplication Authentication Model

Let the number of elements for the *response* of *A* be n×1 and each element in a vector be *k* bits. To blindly retrieve the correct *response*, an attacker needs to take exponentially time complexity $O(2^{k\cdot n})$.

Subsequently, we analyze the value modeling attacks to stochastic logic authentication model.

#### 6.2.6. The Value Modeling Attacks to Stochastic Logic Authentication Model

With the assumption of the strong PUF described in Section 3, the two *responses* generated in the model are mutually independent. Moreover, within one *response*, all individual bits are mutually independent. As a result, an attacker can model the value calculated in step 5 of stochastic logic authentication model as a random variable, which is the transformation of *i* independent binomial trials with probability of success equals to 0.5, where the transformation is the sum of these independent binomial trials. After transformation, value is a random variable with binomial distribution (i,0.5). The probability mass function (PMF) of binomial distribution is ik·0.5k·0.5i−k, where *k* is the number of success, namely the value. The mean of value is 0.5i and for binomial distribution, the P(value=0.5i) has the largest probability compared to other possible points. For an attacker, he/she will always guess the value with largest probability. In other words, the attacker should always guess the value to be 0.5i. For instance, given $i=2^{10}$, P(value=29)=21029·0.529·0.529≈0.0249. Obviously, it is relatively easy for an attacker to break the authentication system with probability 0.025. When the size of $P^{D}$ arises, P(value=0.5i) will decrease. When *i* is large, it is hard to compute the PMF of binomial distribution. However, we can approach to value by using normal distribution, in which the mean is 0.5i and variance is 0.25i. Given $i=10^{8}$, we can approach to value using a random variable with $N(\frac{10^{8}}{2},\frac{10^{8}}{4})$. Using continuity correction, $P\left(value=0.5i\right)=P\left(value=\frac{10^{8}}{2}\right)\approx P\left(49999999.5<X<50000000.5\right)$, where *X* is a random variable with $N(\frac{10^{8}}{2},\frac{10^{8}}{4})$. After standardization, $P\left(value=0.5i\right)=P\left(value=\frac{10^{8}}{2}\right)\approx P\left(-0.0001<Z<0.0001\right)=0.00008$, where *Z* is a random variable with N(0,1).

As the aforementioned descriptions, the cell area of matrix multiplication authentication model entails polynomial growth with respect to the size of $P^{D}$. In contrast, when *i* increases, authentication model based on stochastic logic poses slightly increasing on the size of cell area.

#### 6.2.7. Resistance against Machine-Learning Attacks

The resistance against machine-learning attacks is a characteristic of a PUF representing the difficulty to predict its CRPs using machine-learning techniques [16]. Our approach STT-DPSA used the strong STT-MRAM PUF, which means that the number of CRPs is exponentially in accordance with the number of components whose behavior depends on manufacturing variation of MTJs. The variation of MTJs has been proven by [32,33], in which response cannot be predicted better than random guesses; that is, inter-Hamming distance being equal to 0 and uniformity of response being equal to 0.5. As a result, STT-DPSA can resist against machine-learning attacks.

## 7. Evaluation

To prove the uniqueness of $P^{S}$, we demonstrated the simulation results of $P^{S}$ under different temperatures. In addition, we demonstrated synthetic results of delay of critical path and cell area with different parameters of STT-DPSA based on ASIC cell-based design flow.

The standard cell library used to synthesize the verilog code of proposed model is NanGate FreePDK45 open cell library [34]. Each authentication round in our model STT-DPSA takes 5 clock cycles. In addition, we implemented LFSR [12] as a PRNG in STT-DPSA.

### 7.1. Unpredictability Validation of $P^{S}$

Our STT-MRAM PUF $P^{S}$ circuit was implemented in SPICE model [35] via Monte Carlo simulation with three-dimensional variation of the MTJ shape. The simulation results are shown in Figure 11. The upper sub-figure is the waveform of signal RCLK and the others are waveforms of signal BIT under temperatures in unit of Celsius: 0, 50, and 80 from top to down. Initially, both MTJ A and MTJ B are in the *P* state. We set signal RCLK to Vdd so that BIT is 0. Then we switch signal RCLK to 0, followed by BIT will be pre-charged to the undetermined level. Finally, BIT changes to 0/1 with the corresponding MTJ resistance difference between MTJ A and MTJ B. In Figure 11, the simulation results of signal BIT diverging at different time slots demonstrate that our $P^{S}$ is unpredictable under different temperatures.

As exhibited in Figure 11, *response* bits of $P^{S}$ can be generated within around 30 nanoseconds (ns). Using these *response* bits, the construction of $P^{D}$ can be completed by executing ASIC design flow within limited amount of time. Therefore, with the designs of $P^{D}$ and $P^{S}$, we can realize the initialization phase of STT-DPSA with constant time complexity rather than the traditional PUF-based models that need unrealistic amount of time to generate CRPs of a strong PUF.

### 7.2. Reliability Validation of $P^{S}$

To demonstrate the reliability of $P^{S}$, we validated whether our $P^{S}$ was unchangeable under variations of temperature and Vdd. In Figure 12, the results revealed that the working windows between temperature interval (−35 ${}^{\circ }$C, 110 ${}^{\circ }$C) and Vdd interval (0.95 V, 1.16 V) for $P^{S}$ are reliable.

### 7.3. Effect of the Parameter *l*

We varied the length *l* of $P^{D}$ when fixed the length i=64 of input bits to the $P^{D}$ and let *k* be equal to 8. Figure 13 and Figure 14 demonstrated the delay of critical path and the cell area, respectively, in which *l* posed linear growth on the delay of critical path and the cell area. Obviously, the delay of critical path and the cell area are about 8 ns and 2.17 um*um, respectively, for both of practical and theoretical evaluations at l=4.

### 7.4. Effect of the Parameter *n*

We exhibited the effect of the parameter *n* on the delay of critical path and the cell area as l=2 and k=8 in Figure 15 and Figure 16. Because we suppose that the analysis presented in Theorem 2 is the naivest hardware architecture, different hardware architectures in practice would affect the evaluated results. In our hardware platform, the evaluated delay of critical path and cell area are lower than the theoretical estimation. Moreover, Figure 15 and Figure 16 indicated that our model posed linearly growth overhead on delay of critical path and posed polynomially growth overhead on cell area while varying the parameter *n*.

### 7.5. Effect of the Parameter *k*

In this section, effects of the parameter *k* on the delay of critical path and the cell area as l=2 are shown in Figure 17 and Figure 18. Observing from Figure 17 and Figure 18, the parameter *k* posed linearly growth on the delay of critical path and the cell area. At k=16, the evaluated delay of critical path and cell area are about 8.5 ns and 3.5 um*um, respectively.

### 7.6. Execution Time of Authentication

We evaluated the execution time of authenticating a client in our proposed model compared to a brute-force method launched by an attacker while varying the parameters *k* and *n*. The evaluated results are shown in Figure 19 and Figure 20. As previous discussions, the most effective attack against our proposed model is to blindly guess the *response* of the matrix *A*. Consequently, the security of proposed authentication model is $O(2^{k\cdot n})$. As a result, we can create giant time execution gap between the brute-force method and our model. For instance, the attacker will take about computational complexity $O(2^{256})$ to compromise STT-DPSA under the settings of n=32, k=8, i=o=1024. This security level is equivalent to the AES-256 cryptography system.

### 7.7. Evaluation of Stochastic Logic Authentication Model

In this section, we set the size of $P^{D}$ to i=o=64 and indicated the parameters *n*, *k*, and *l* to 8, 8, and 2, respectively. Under this assumption, we used theoretical equation derived in Section 1 to estimate the cell area and the delay of critical path for matrix multiplication and stochastic logic authentication models. Table 2 demonstrated the comparison results. Considering hardware security level or cell area, we alternatively build matrix multiplication or stochastic logic operation for our authentication model. In other words, if we would like to reserve high security for authentication, matrix multiplication model can be used while sacrificing area overhead and delay of critical path; otherwise, stochastic logic model can be used to improve performance of cell area and delay of critical path for STT-DPSA.

### 7.8. Comparisons

STT-DPSA was compared with DBF [14], DBF plus Arbiter [15], PUFSec [24], and Matched PUF [13] in driving device authentication, as shown in Table 3. In our evaluations, STT-DPSA and Matched PUF used pure hardware approach while DBF, DBF plus Arbiter, and PUFSec used hardware-software co-design methodology. For fairly comparison, we set the security level at $O(2^{64})$ for each method and compared them in terms of the cell area, execution time to finish one authentication round, and memory usages. Notably, our evaluation was under the ASIC cell-based design flow using NanGate FreePDK45 open cell library [34].

STT-DPSA outperforms DBF, DBF plus Arbiter and PUFSec in terms of execution time and memory usage while sacrificing cell area. STT-DPSA only spent 31.45 ns for performing one round of authentication while DBF, DBF plus Arbiter and PUFSec cost 0.03, 0.03 and $5.15\times 10^{-10}$ in terms of seconds, respectively. For the memory usage, DBF and DBF plus Arbiter and PUFSec all cost unrealistic memory usage while STT-DPSA did not consume memory usages due to its pure hardware implementation. Comparing STT-DPSA with Matched PUF, Matched PUF outperforms STT-DPSA in terms of cell area, execution time but it is extremely susceptible to environment condition.

### 7.9. Implementation of STT-DPSA

Finally, STT-DPSA was implemented with parameters n=32, i=o=1024, k=8, and l=2 using FPGA design flow. The FPGA device that we used is EP4CE115F29C7, which belongs to Cyclone IV E FPGA family of Intel. Under this setting of parameters, an attacker needs to take time complexity $O(2^{256})$ to compromise STT-DPSA. We also evaluated STT-DPSA using Synopsys design compiler with TSMC 0.18 um process. Table 4 is the statistics of STT-DPSA implemented on the FPGA device and Table 5 is the specification of the TSMC 0.18 um process implementation for STT-DPSA. They demonstrated the feasibility of STT-DPSA.

## 8. Conclusions

We presented a novel authentication model based on a strong digital PUF for IoT devices. In our implementation, we generated a strong digital PUF using STT-MRAM PUF to configure LUTs in the strong digital PUF. By deploying the strong digital PUF configured by STT-MRAM PUF on the client and verifier, the execution time of doing one authentication round and the cell area is efficient. At the same time, our proposed authentication model can resilient against a series of man-in-the-middle attacks and brute-force attacks. The security of STT-DPSA was analyzed in the theoretical estimation. Moreover, we validated the cell area, the delay of critical path, and the execution time of one authentication round in experiments. Finally, we realized our proposed model using FPGA design flow to prove the feasibility of STT-DPSA.

## Figures and Tables

**Figure 1 micromachines-11-00502-f001:**
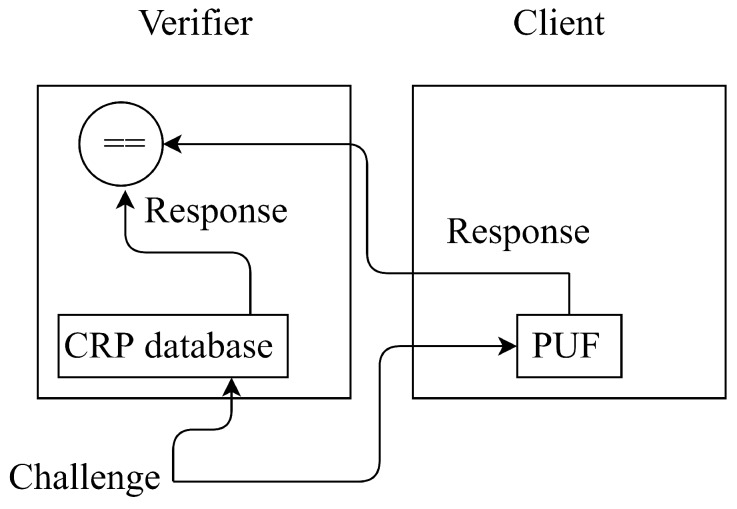
Typical strong PUF-based authentication model.

**Figure 2 micromachines-11-00502-f002:**
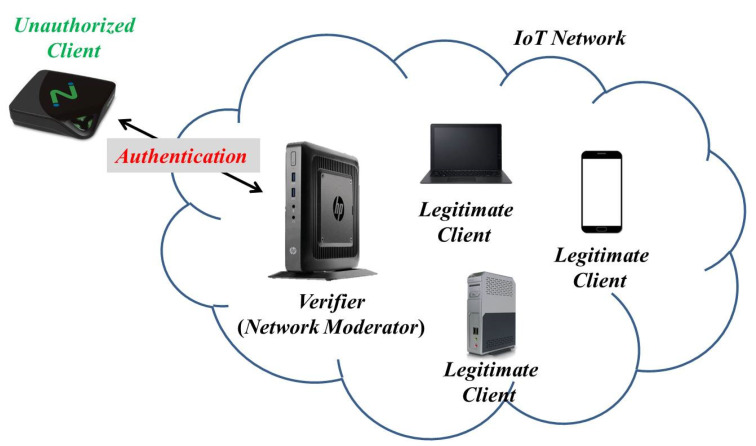
System model.

**Figure 3 micromachines-11-00502-f003:**
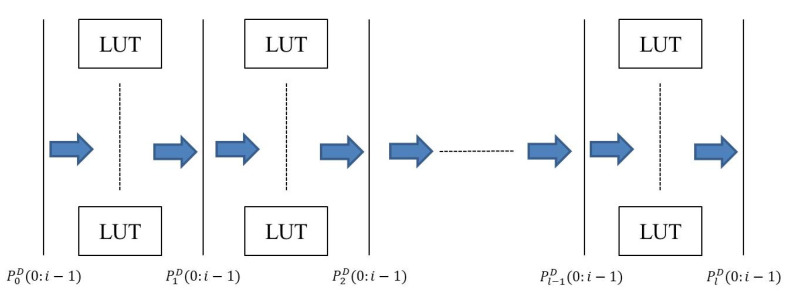
Structure of $P^{D}$.

**Figure 4 micromachines-11-00502-f004:**
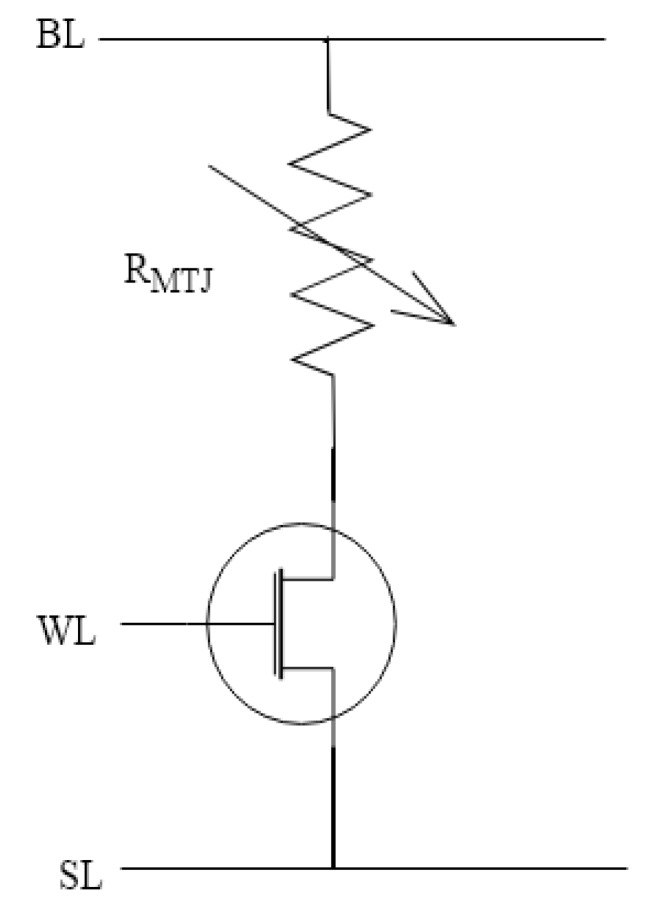
A cell of STT-MRAM.

**Figure 5 micromachines-11-00502-f005:**
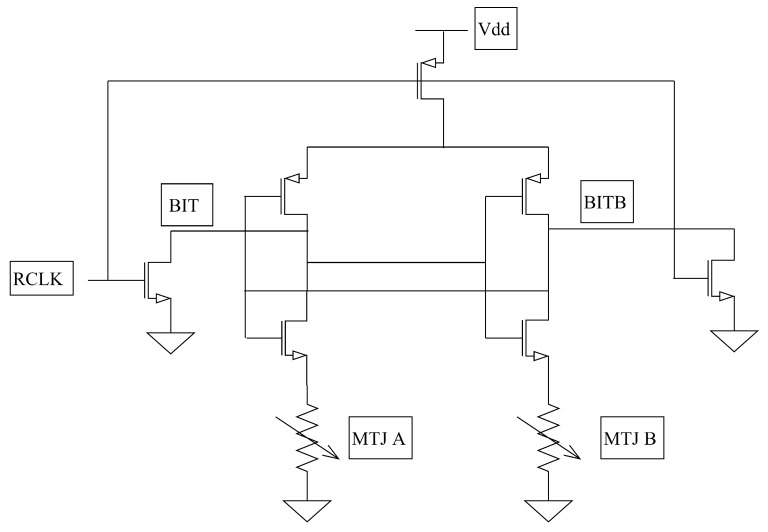
Design of $P^{S}$.

**Figure 6 micromachines-11-00502-f006:**
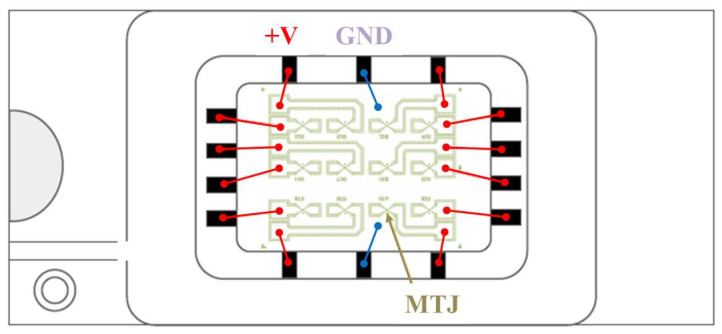
Dual in-line package of our MTJs.

**Figure 7 micromachines-11-00502-f007:**
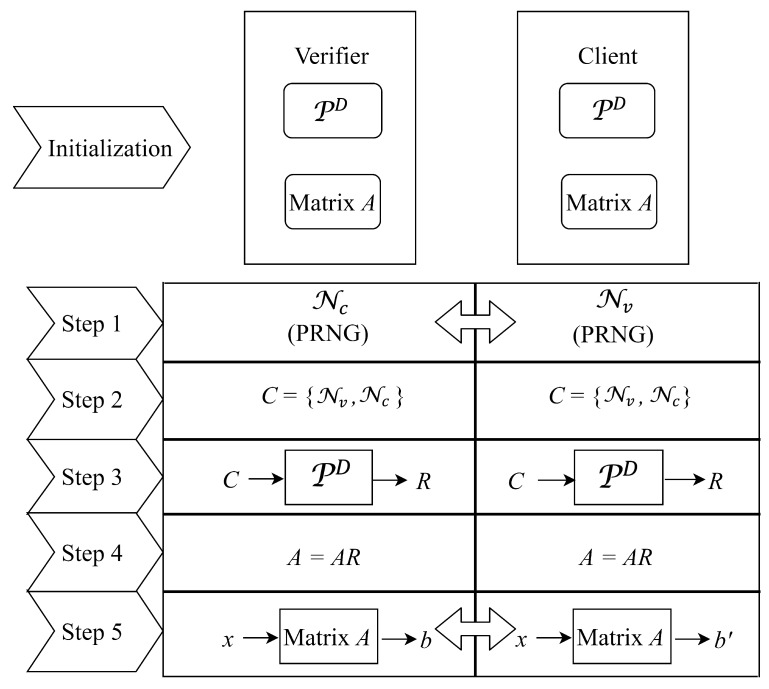
Steps of authentication model.

**Figure 8 micromachines-11-00502-f008:**
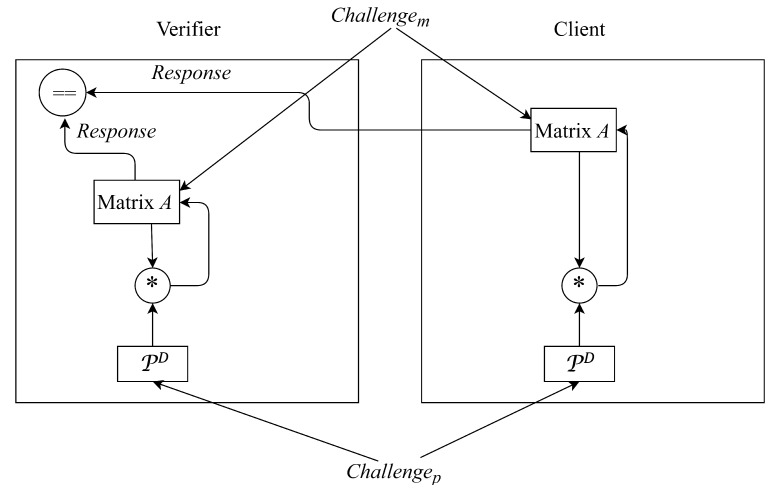
Flow chart of authentication model based on matrix multiplication.

**Figure 9 micromachines-11-00502-f009:**
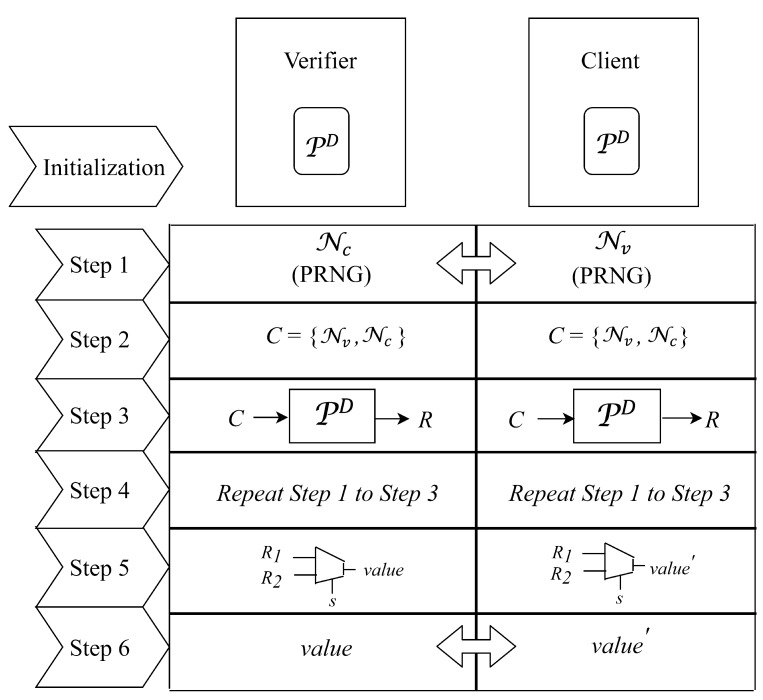
Steps of stochastic logic-based authentication model.

**Figure 10 micromachines-11-00502-f010:**
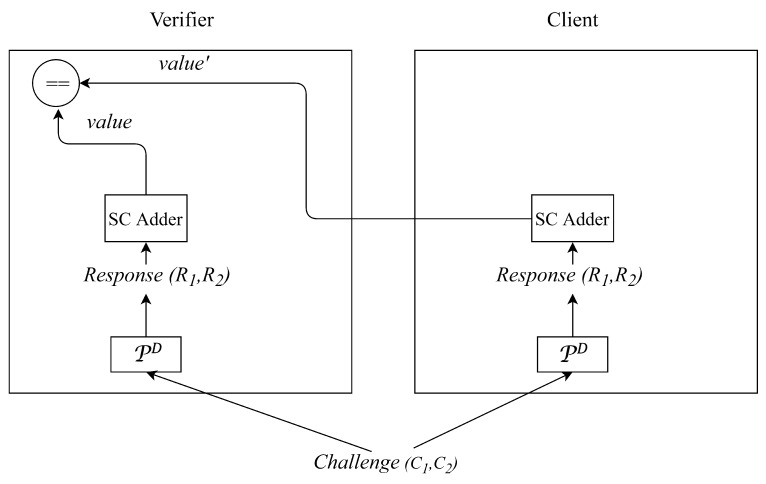
Flow chart of the authentication model based on stochastic logic (SC).

**Figure 11 micromachines-11-00502-f011:**
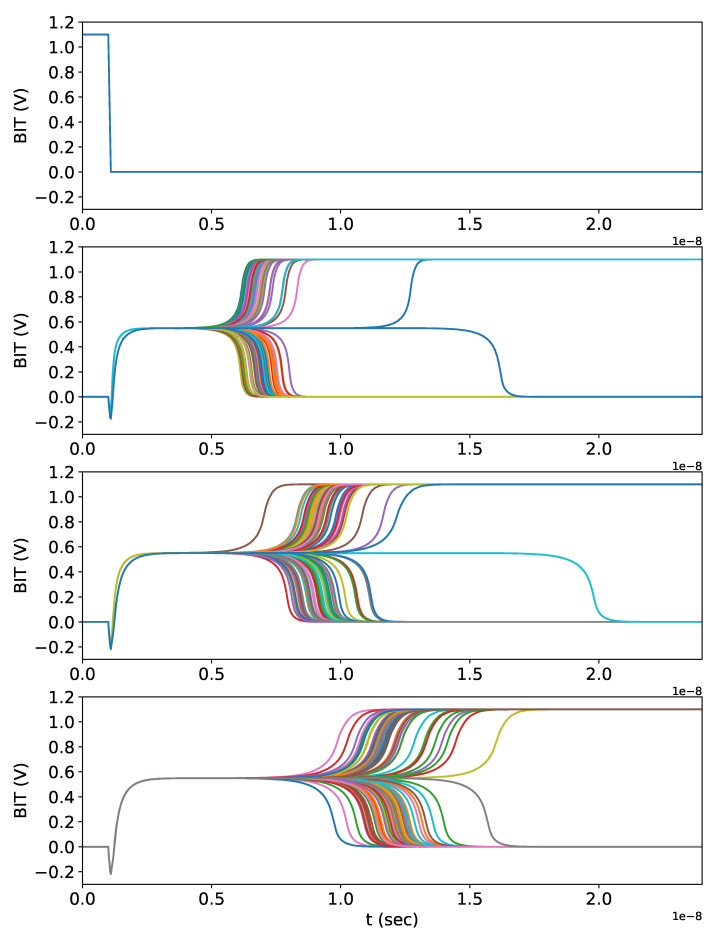
The uniqueness validation of $P^{S}$.

**Figure 12 micromachines-11-00502-f012:**
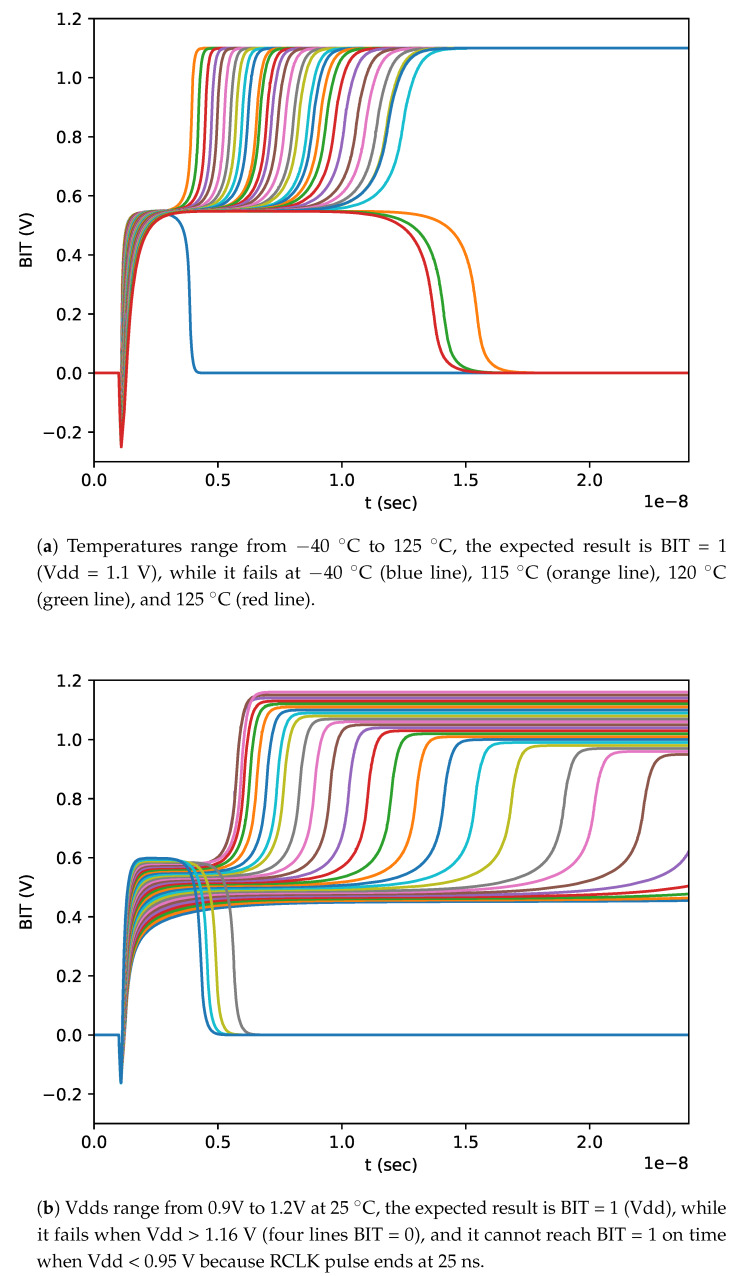
Reliability validation of $P^{S}$ under two MTJs with a bit resistance-area product (RA) difference, MTJ A: 5 Ω-μm${}^{2}$ and MTJ B: 5.2 Ω-μm${}^{2}$.

**Figure 13 micromachines-11-00502-f013:**
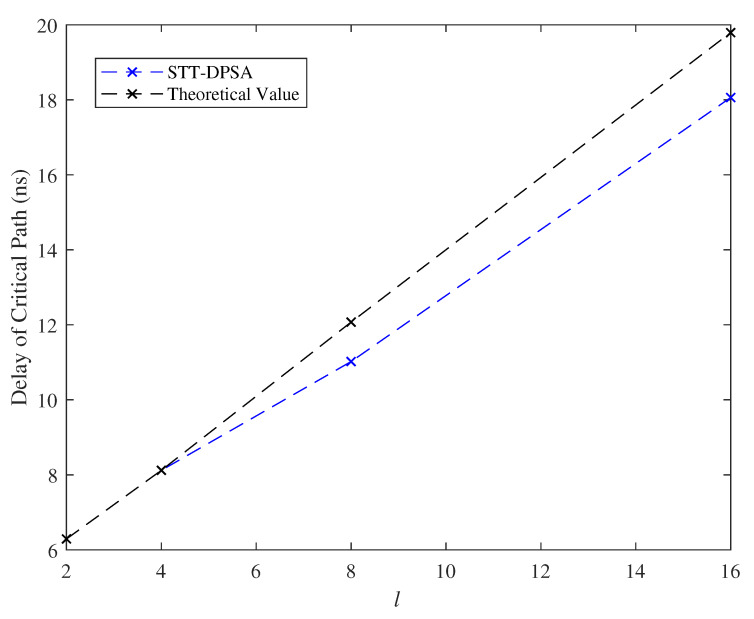
Effect of the parameter *l* with respect to the delay of critical path.

**Figure 14 micromachines-11-00502-f014:**
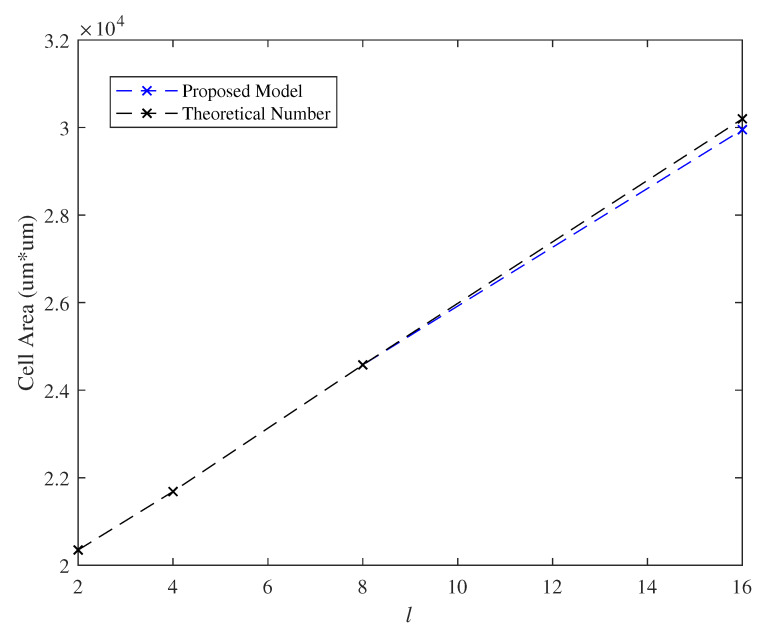
Effect of the parameter *l* with respect to the cell area.

**Figure 15 micromachines-11-00502-f015:**
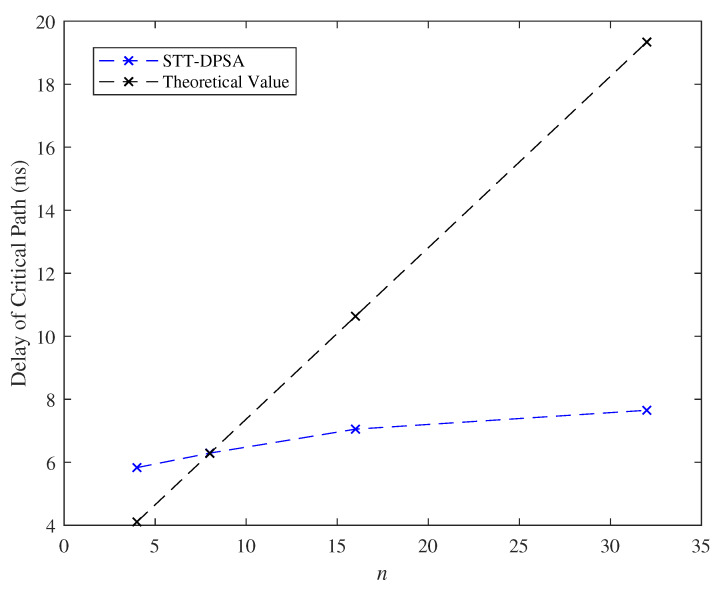
Effect of the parameter *n* with respect to the delay of critical path.

**Figure 16 micromachines-11-00502-f016:**
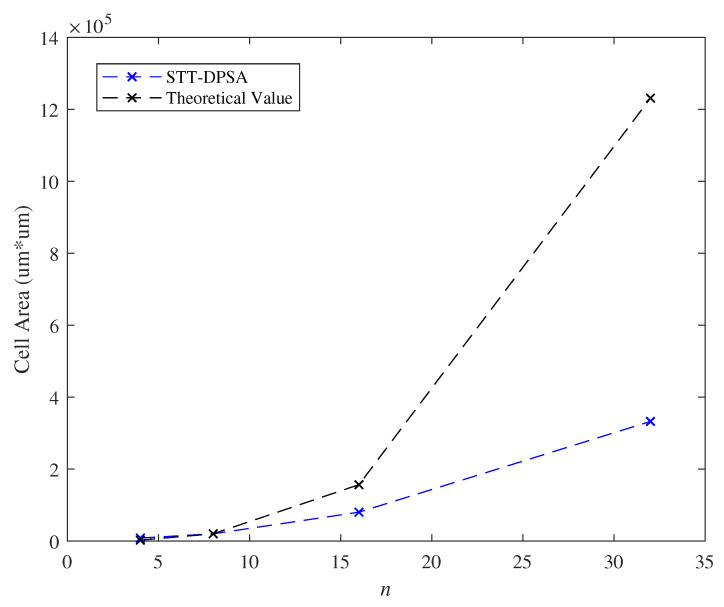
Effect of the parameter *n* with respect to the cell area.

**Figure 17 micromachines-11-00502-f017:**
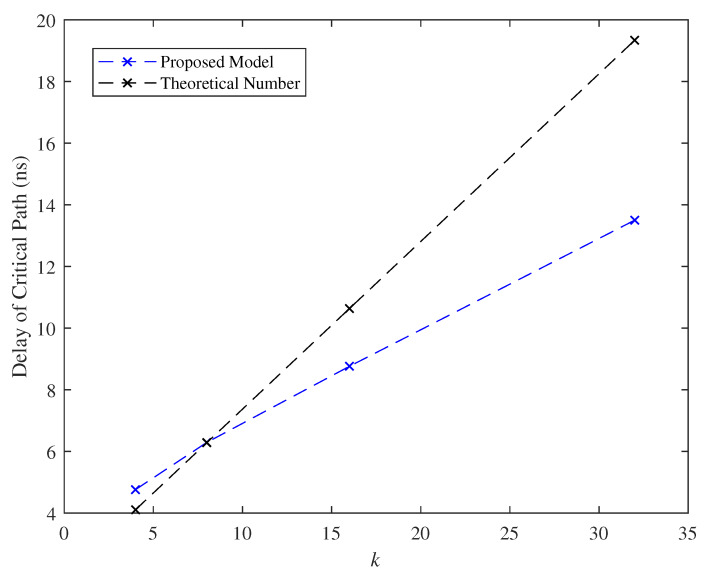
Effect of the parameter *k* with respect to the delay of critical path.

**Figure 18 micromachines-11-00502-f018:**
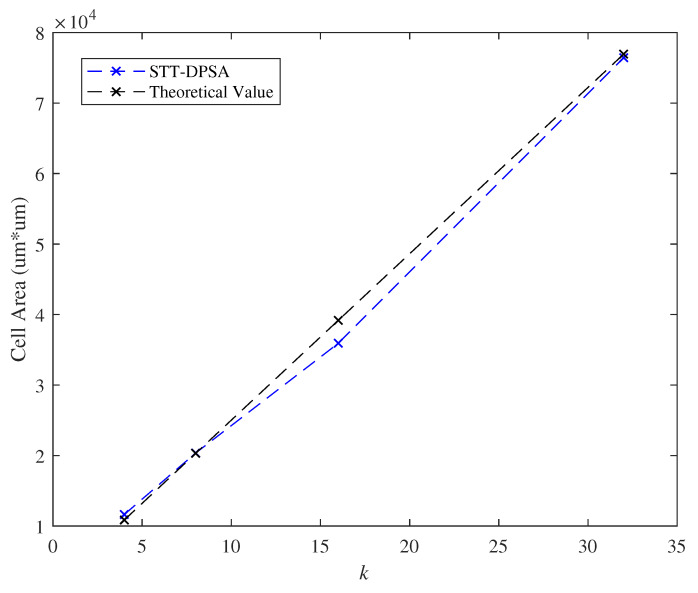
Effect of the parameter *k* with respect to the cell area.

**Figure 19 micromachines-11-00502-f019:**
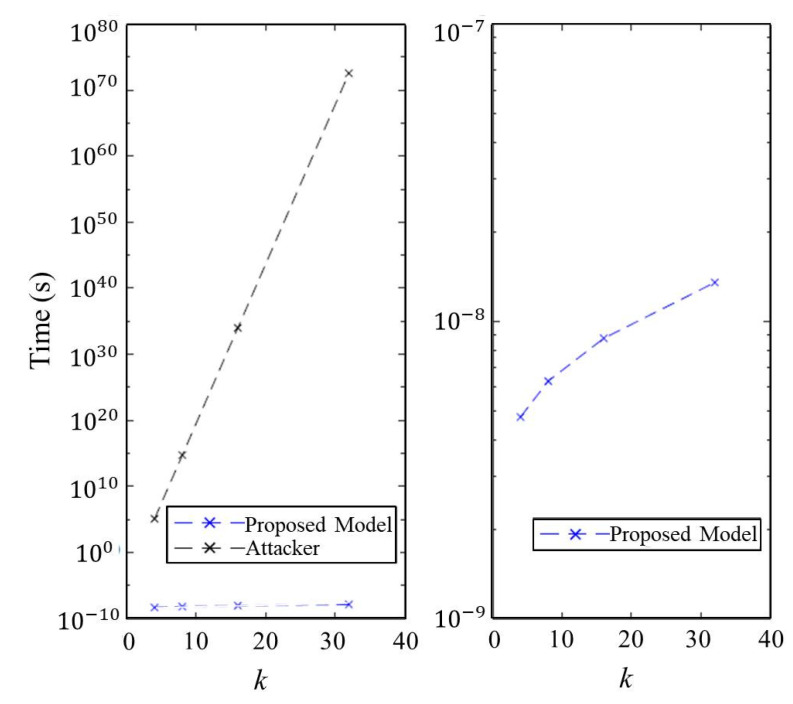
Execution time of an attacker and STT-DPSA to do authentication under n=8 and l=2 while varying the parameter *k*.

**Figure 20 micromachines-11-00502-f020:**
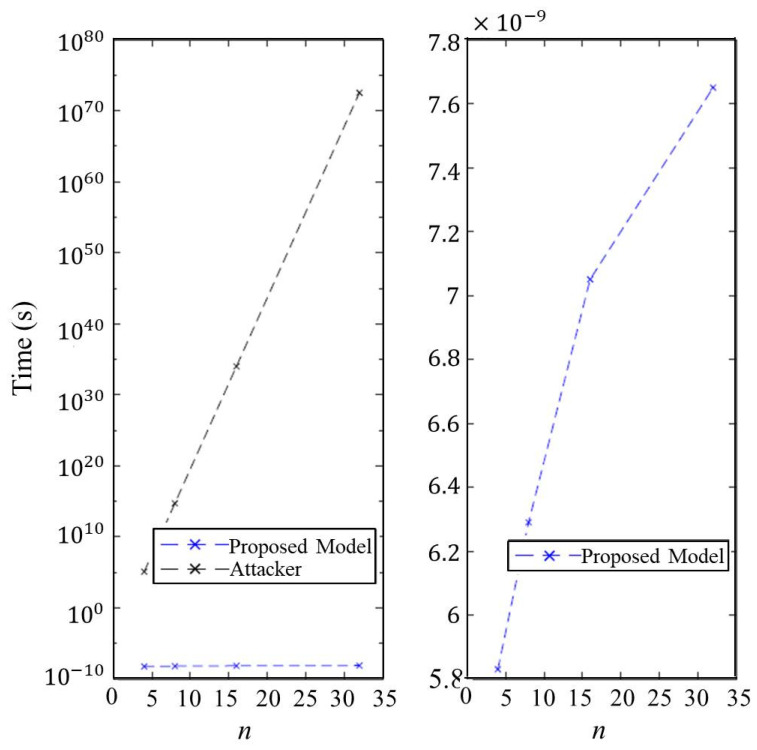
Execution time of an attacker and STT-DPSA to do authentication under k=8 and l=2 while varying the parameter *n*.

**Table 1 micromachines-11-00502-t001:** Comparisons.

Methods	Computationa Complexity	Computational Complexity	Susceptibility	Initialization Time	Resource Complexity
	(Verifier Side)	(Client Side)			
**PUFSec [24]**	Exponential	Exponential	Yes	Linear	Exponential
**Matched PPUF [13]**	Linear	Linear	Yes	No	Constant
**DBF [14]**	Polynomial	Linear	No	Exponential	Exponential
**DBF with Arbiter PUF [15]**	Polynomial	Linear	Exponential	Yes	Exponential
**STT-DPSA (our method)**	Linear	Linear	No	Linear	Polynomial

**Table 2 micromachines-11-00502-t002:** Comparison results between stochastic logic version and matrix multiplication version.

	Cell Area (μm*μm)	Delay of Critical Path (ns)
Matrix Multiplication Version	20,349	31.45
Stochastic Logic Version	4953	17.25

**Table 3 micromachines-11-00502-t003:** Comparison results.

Method	Cell Area (μm*μm)	Execution Time (s)	Memory Usage (bits)
STT-DPSA	20,349	$31.45\times 10^{-9}$	0
DBF [14]	835	0.03	$2^{64}$
DBF with Arbiter PUF [15]	930	0.03	$2^{64}$
PUFSec [24]	2245	$5.15\times 10^{-10}$	$2^{56}$
Matched PUF [13]	5376	$5\times 10^{-9}$	0

**Table 4 micromachines-11-00502-t004:** Statistics of the FPGA implementation.

Parameter Settings	n=32, i=o=1024, k=8, and l=2
FPGA Device	Intel Cyclone IV E EP4CE115F29C7
Total Logic Elements	68108/114480 (59%)
Total Registers	3080
Clock Frequency	$10^{7}$ Hz

**Table 5 micromachines-11-00502-t005:** Specification of the TSMC 0.18 um process implementation for STT-DPSA.

Supply Voltage (V)	Cell Area (um${}^{2}$)	Power @ 50 MHz (mW)
1.8	20,349	1.2278

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
