# Peer review of "STT-DPSA: Digital PUF-Based Secure Authentication Using STT-MRAM for the Internet of Things"

_micromachines, 2020, doi:10.3390/mi11050502_

Round 1
Reviewer 1 Report
This paper proposes a new digital PUF (STT-DPSA) for authentication. The writing of this paper is satisfactory. The novelty of this paper is sufficient. However, there are some key points missing in this paper. Since the authors used PUF and PRNG for authentication, they need to offer more information to discuss the corresponding robustness under machine learning attacks. Moreover, the authors should provide the performance analysis of the proposed PUF, such as reliability versus temperature and reliability versus supply voltage. In table 3 of this paper, the area overhead of the proposed STT-DPSA is extremely high as compared to other PUFs. Is there any way to mitigate this high area overhead? It is better for the authors to give more information to discuss this area overhead issue. Finally, I would like to see the comparison between the proposed work and the below references.
[1]Tanaka, S. Bian, M. Hiromoto, and T, Sato, “Coin Flipping PUF: A Novel PUF With Improved Resistance Against Machine Learning Attacks,” IEEE Transactions on Circuits and Systems II: Express Briefs, vol. 65, no. 5, pp. 602-606, May 2018.
[2]. W. Yu, Y. Wen, S. Köse, and J. Chen, “Exploiting Multi-Phase On-Chip Voltage Regulators as Strong PUF Primitives for Securing IoT,” Journal of Electronic Testing: Theory and Applications (Springer), vol. 34, no. 5, pp. 587-598, October 2018.
[3]. M. Rahman, A. Hosey, Z. Guo, J. Carroll, D. Forte, and M. Tehranipoor, “Systematic correlation and cell neighborhood analysis of SRAM PUF for robust and unique key generation,” Journal of Hardware and Systems Security, vol. 1, no. 2, pp. 137-155, June 2017.
[4]. W. Yu and J. Chen, “Masked AES PUF: a new PUF against hybrid SCA/MLAs,” IET Electronics Letters, vol. 54, no. 10, pp. 618-620, May 2018.
[5]. R. Govindaraj, S. Ghosh, and S. Katkoori, “Design, Analysis and Application of Embedded Resistive RAM based Strong Arbiter PUF,” IEEE Transactions on Dependable and Secure Computing, https://ieeexplore.ieee.org/abstract/document/8443101
[6]. Y. Wen and W. Yu, “Machine learning-resistant pseudo-random number generator,” IET Electronics Letters, vol. 55, no. 9, pp. 515-517, May 2019.
Author Response
REVISION STATEMENT
Manuscript ID: micromachines-762388 entitled “STT-DPSA: Digital PUF-based Secure Authentication Using STT-MRAM for the Internet of Things”
We would like to express our gratefulness to the reviewers for their precious comments, which were very constructive and helpful in clarifying the manuscript. We also thank the editor for the effort in handling our manuscript.
We have taken reviewers’ comments into careful consideration to revise our manuscript. Our responses to reviewers’ comments are described in detail as follows.
Please fine our replies to the reviewer's comments in the attachment.

Reviewer 2 Report
This manuscript proposes a digital PUF-based authentication model using STT-MRAM PUF in the IoT paradigm. The paper is well structured and the results are supported by correct simulations.
- You should give specific errors in Abstract
- Don't give the solution of your proposal at the end of the introduction
- Develop all references in related work, you just add them in series without explaining and comparing them
- The contributions must be explained at the end of the introduction
- Low-quality figures, improve them
- Cannot read letters in fig 6
- remove code in line 212
- We already know what stochastic adder and multiplier are remove figures 9 and 10
- Increase size text in figures and plots
Author Response

(The authors gave the same response as above.)

Round 2
Reviewer 1 Report
The paper has been significantly improved. I recommend acceptance for the present form. Just a tiny concern, can the authors check the 5th row of Table 3? Is the execution time of PUFsec[24] is 5.15*10^(10) or 5.15*10^(-10)?